# Maximization of Open Hospital Capacity under Shortage of SARS-CoV-2 Vaccines—An Open Access, Stochastic Simulation Tool

**DOI:** 10.3390/vaccines9060546

**Published:** 2021-05-22

**Authors:** Wolfram A. Bosbach, Martin Heinrich, Rainer Kolisch, Christian Heiss

**Affiliations:** 1Experimental Trauma Surgery, Justus Liebig University of Giessen, 35392 Giessen, Germany; martin.heinrich@chiru.med.uni-giessen.de (M.H.); christian.heiss@chiru.med.uni-giessen.de (C.H.); 2Department of Trauma, Hand, and Reconstructive Surgery, University Hospital of Giessen, 35385 Giessen, Germany; 3Covid-19 Emergency Taskforce, University Hospital of Giessen, 35385 Giessen, Germany; 4TUM School of Management, Technical University of Munich, 80333 Munich, Germany; rainer.kolisch@tum.de

**Keywords:** SARS-CoV-2 vaccine shortage, hospital vaccine rollout, hospital management

## Abstract

*Motive.* The Covid-19 pandemic has led to the novel situation that hospitals must prioritize staff for a vaccine rollout while there is acute shortage of the vaccine. In spite of the availability of guidelines from state agencies, there is partial confusion about what an optimal rollout plan is. This study investigates effects in a hospital model under different rollout schemes. *Methods.* A simulation model is implemented in VBA, and is studied for parameter variation in a predefined hospital setting. The implemented code is available as open access supplement. *Main results.* A rollout scheme assigning vaccine doses to staff primarily by staff’s pathogen exposure maximizes the predicted open hospital capacity when compared to a rollout based on a purely hierarchical prioritization. The effect increases under resource scarcity and greater disease activity. Nursing staff benefits most from an exposure focused rollout. *Conclusions*. The model employs SARS-CoV-2 parameters; nonetheless, effects observable in the model are transferable to other infectious diseases. Necessary future prioritization plans need to consider pathogen characteristics and social factors.

## 1. Introduction

With the availability of Covid-19 vaccinations [1,2], hospital operations management has faced worldwide the new situation that a vaccine rollout scheme for hospital staff had to be implemented. Due to scarcity of vaccine doses, prioritization decisions have to be made during this rollout. This study makes a contribution to this topic by investigating consequences from different rollout schemes. The study’s focus is the maximization of open hospital capacity which is assumed to maximize patient benefit.

Research studies from the pre-Corona age exist. Their focus was to prove benefit from vaccination policies [3,4,5]. Shortage of vaccine supply was not a particular focus topic; it was known e.g., for yellow fever [6]. Instead, rather an oversupply and mandatory vaccination were discussed [7]. The early 2020 vaccination framework of the World Health Organization (WHO) already included aspects of prioritization decisions for a vaccination rollout against the severe acute respiratory syndrome coronavirus 2 (SARS-CoV-2) [8]. The general logistics of the large scale Covid-19 vaccine rollout currently taking place require vaccine doses, vaccinating staff administering the doses, and a process to assign them to patients [9]. Rollout recommendations or deployment plans have been developed and published by state agencies [10,11], also with details on the process of vaccine recommendation decisions [12].

There is a general agreement that vaccines should be used to their best potential to curb the pandemic’s consequences, in particular during the early vaccine rollout phase when shortage of vaccine supply prohibits an immediate full rollout to the entire population. Decision making under shortage of resources is known from other important medical supplies in connection with the current pandemic [13,14]. Insights into prioritization decisions and their consequences have been published for studies considering the entire population [15,16,17,18,19]. Delaying a second dose if required by the drug regime has been discussed as an option to reduce vaccine shortage [20,21]. In spite of the available material and dedicated rollout recommendations, e.g., [10,11], decision making about rollouts to hospital staff can still prove to be controversial. The weighing in of factors such as hierarchical importance [22,23] or student status [24] can lead to differences in the rollout scheme and prioritization among hospital staff. The rollout in the United Kingdom so far has been successful [25]. Subgroups of a population such as those economically worse off [26] or minorities [27] can require special attention, however. In addition, nursing staff, who by the nature of their work are in close contact to patients, must not be forgotten in rollout schemes and prioritization [28].

In this presented study here, a model hospital is simulated and its capacity is calculated. The hospital’s capacity is limited by available staff. Staff are exposed to the pathogen and are vaccinated after two different rollout schemes which are compared to each other. The first rollout scheme assigns vaccine doses in **hierarchical order top down**. It relates to arguments made in the past [22,23] that higher-ranking staff are fewer in number, more important as individuals for the functioning of the hospital, and that they are of greater age, which can increase the transmission probability of a pathogen. The alternative rollout scheme first prioritizes between hospital units by **exposure to the pathogen**. Accidents & emergencies (A&E), where exposure to nontested outpatients occurs, is assigned vaccine doses first and hospital wards follow behind. On each unit, vaccines are assigned in hierarchical order.

The model is implemented as Visual Basic for Application (VBA) macro. The macro and its embedded version in Microsoft-Excel (Microsoft Corporation, Redmond, WA, USA) are available as open access supplement 1 and 2 under the GNU’s Not Unix (GNU) General Public License version 3, or any later version. We hope that the open access macro code (Appendix A) will further exchange, and will make accessibility to the study’s work easier. A similar project with results for an entire population, split by age groups, is published in [18], and pre-Corona work about hospital staff in [3,4,7]. The study’s methodology uses an algorithm similar to known work from hospital capacity planning [29]. Staff and vaccines are modelled as streams, similar to [30]. It applies novel insights into the epidemiological spreading of SARS-CoV-2 [31,32], and factors influencing disease spreading such as age [33,34].

The study’s parameters are set to values typical for the current SARS-CoV-2. However, the methodology and the implemented model are equally applicable to other infectious diseases. The simulated hospital and affected staff are specified in Section 2.1 and Section 2.2. Staff structure is a pyramid. This hierarchy form is found worldwide as typical human resources (HR) structure in hospitals. Staff further up in the hierarchy is of greater age. Larger simulations are possible in the future. The investigated effects are independent of hospital size. In Section 4, we discuss observable trends and predictions for future scenarios.

Although mostly gone unnoticed in Europe and North America, the list of major infectious disease outbreaks in e.g., Hong Kong during the 25 years before the ongoing SARS-CoV-2 pandemic included avian flu H5N1 in 1997, SARS in 2003, swine flu 2009, and avian flu H7N9 in 2013 [35]. Hence, a vaccine rollout in the general population and amongst hospital staff against an infectious disease might be needed again in the future.

## 2. Methodology and Model Implementation

For the present study, a hypothetical hospital is modelled as it can be found around the world. Staff on higher hierarchy levels are fewer in number, of greater age, and of greater individual importance for the functioning of the hospital organization. Influence of model parameters is discussed in the results section and in the conclusions section.

### 2.1. Hospital Structure and Staff Reserve

The hospital is structured in four units. Staff is coming to work every day and is simulated for a variation of initially available staff reserve. Staff size and staff requirements are chosen to reflect a typical hospital situation.

**Hospital staff.** The hospital staff consists of doctors and nurses. Nurses are not substructured further. Doctors are each assigned to the rank of physician, senior doctor, executive senior, or chief. The model assumes that any staff member is qualified to work on any hospital unit. Table 1 summarizes the staff numbers initially available at *t* = 0 for the simulated base case and the scenarios of staff reserve variation.

**Hospital units.** The hospital is structured in four units. It is assumed that the hospital operates under the specialty of general internal medicine. A&E receives outpatients. In-patients are treated on wards 1–3. The daily numbers of staff required for 100%/50% operations of each unit are shown in Table 2. Nurses and physicians are required to be present. The director, executive seniors, and senior doctors provide background service. The chief and executive seniors are expected during staff shortage also to work in the role of senior doctor; they are downward compatible. The specified staff requirements reflect a typical contemporary hospital organization. The total and relative numbers are chosen to imitate a general medicine or surgery clinic.

Hospital capacity Γ is calculated as average of the open status of A&E, and ward 1–3 in Equation (1).
(1)Γ=∑iγi4

Γ: relative open hospital capacity, *γ*: hospital unit’s open status, $i\in \left[A\&E,{\;\mathrm{ward}}_{1},{\mathrm{ward}}_{2},{\mathrm{ward}}_{3}\right]$.

### 2.2. Age Distribution

It is known that there can be an age factor in the epidemiology of a pathogen. SARS-CoV-2 is no exception in that regard [33,34]. For each staff member, we assign an age based on a uniform distribution. Table 3 gives the bounds of the distribution for the different staff roles. The impact of age on infection events is discussed further in Section 2.4. Age boundaries vary between different countries; e.g., there is no global uniform retirement age. The chosen age distribution reflects a typical hospital setting of our time.

### 2.3. Vaccination Rollout

The model compares two rollout schemes. In each case, the rollout rate *v_S_* defines the number of vaccines per day available to the hospital. Staff are vaccinated, unless they are symptomatic, or already vaccinated. The model is implemented for a vaccine which applies only one dose per person.

**Top down vaccine rollout.** In the case of the top down rollout scheme, the model assigns doses hierarchically and moves from the top down, starting with the chief. Senior executives, senior doctors, and physicians follow. Nurses are assigned doses last.

**Exposure focused vaccine rollout.** The exposure focused rollout scheme prioritizes staff by their exposure to the pathogen during their hospital shifts. Due to symptomatic and nonsymptomatic outpatients walking in at A&E, A&E staff’s exposure is assumed greater than exposure of staff on the wards 1–3. A&E staff are vaccinated first; staff on wards 1–3 follow unit by unit in chronological order. On A&E and wards 1–3, again a hierarchical vaccine assignment is implemented.

### 2.4. Disease Status and Pathogen Transmission

The model calculates each day for each staff member a pathogen transmission probability and the individual disease status.

#### 2.4.1. Staff Member’s Disease Status

The disease status of a staff member can be noninfected, infected and nonsymptomatic, or infected and symptomatic. After pathogen transmission, a staff member is infected, nonsymptomatic and continues working. After the passing of the incubation time $t_{inc}$ [36], the staff member becomes symptomatic and remains off duty for the recovery duration of $t_{rec}$. Whether reinfections with SARS-CoV-2 are possible and by what probability is being investigated at the moment [37]. The model assumes that staff can reinfect unless vaccinated. For simplification of disease behavior, staff being vaccinated during $t_{inc}$ are assumed not to be infected.

#### 2.4.2. Pathogen Transmission

The model assumes that the pathogen transmits from infected human to noninfected human. The model assumes that vaccinated staff are immune to the pathogen and are not infectious for other staff members. For each working day, a probability Π is calculated for which a staff member’s status is changed to infected. This probability considers the level of exposure, $R_{S}$ as used in epidemiology as reproductive number [31], the age [33,34], and the reduced patient contact of greater hierarchy ranks. Equations (2) and (3) below detail the calculation of the probabilities $\Pi_{A\&E}$ and $\Pi_{ward1-3}$. Other nonhospital healthcare settings as found in private practice, nursing homes, or at home care services can be simulated alike, based on their specific pathogen exposure patterns.

**Pathogen exposure on A&E and wards 1–3.** Exposure quantifies in the model the contact to infected humans. On A&E, staff comes into contact with outpatients who walk in and are positive with probability rate $p_{pos}$. The model assumes that on A&E each staff member has contact with 20 patients during one shift. The model adds the number of infected, nonsymptomatic colleagues ${\bar{sym}}_{A\&E}$ on A&E who come to work on that day and increase pathogen exposure to other staff members.

On wards 1–3, outside A&E, the model assumes that all patients are tested and, if positive, isolated so that patients cause no pathogen exposure to staff. The model adds again the number of infected, nonsymptomatic colleagues $\bar{sym}$ who come to work on that day to wards 1–3. A base probability of 1/100 is added. By this, the model accounts for the fact that pathogen intake on wards 1–3 vanishes compared to intake on A&E but that it is not nil.

**Infection risk *Π* due to exposure.** Infection risk due to exposure is modelled according to research insights about reproduction numbers found in [31,32] for SARS-CoV-2. These studies have provided probabilities by which an individual can expect to be infected after a specified exposure event (specified by parameters such as pathogen emission rate, breathing activity, aerosol concentration, or duration of exposure). In this context, the parameter $R_{S}$ (pathogen characteristic reproduction number) is used. $R_{S}$ is defined as the number of people infected by one index patient. This model now uses the values for $R_{S}$ of [31], multiplies by the number of positive contacts, and divides it by factor five for obtaining the staff member’s personal risk Π. The reduction by a factor of five is based on the assumption that the relevant index patient has contact to five members of staff.

**Amendment for age and reduced patient contact of greater hierarchy ranks.** It is known that social status and age can influence the spreading of infectious diseases [24,26,27]. The model acknowledges this and considers two additional amendment factors, which are both multiplied on the infection risk Π.

The age factor $T_{S}$ is a number greater or equal 1 and scales linearly between the staff age of 18 and 65, assigned for each staff in Section 2.2. Values are set to the magnitude as known so far for SARS-CoV-2 [33,34].

In modern hospitals, higher hierarchy ranks have typically less patient contact compared to junior doctors or nurses. The model considers this by a factor $H_{S}$ which is set to a value greater or equal one. The infection risk for the hospital’s chief and executive seniors is divided by $H_{S}$.
(2)$$\Pi_{A\&E}=\left(20\;p_{pos}+{\bar{sym}}_{A\&E}\right)\frac{R_{S}}{5}\;T_{S}/H_{S}$$

Π: infection risk, $p_{pos}$: positive rate of patients on A&E unit, ${\bar{sym}}_{A\&E}$: number of positive but nonsymptomatic staff on A&E, $R_{S}$: reproduction number, $T_{S}$: age factor, $H_{S}$: hierarchy factor.
(3)$$\Pi_{i}=\left(\frac{1}{100}+{\bar{sym}}_{i}\right)\;\frac{R_{S}}{5}\;T_{S}/H_{S}$$

$i\in \left[{\mathrm{ward}}_{1},{\mathrm{ward}}_{2},{\mathrm{ward}}_{3}\right]$.

### 2.5. Assignment of Staff to Units and Open/Closed-Status Definition

Staff who is not symptomatic is assigned to work on the hospital’s units. Available staff is by first priority assigned to A&E; followed by wards in ascending ward number. Symptomatic staff is considered off duty.

Open/closed status of a unit is defined for each day by the model according to staff requirements of Table 2.

### 2.6. VBA Implementation

The model has been implemented in VBA. It calculates for a period of 100 days the disease spreading under different scenarios. Each scenario is run *n* = 500 times to account for statistical unevenness of random number function of e.g., age distribution (Section 2.2) or pathogen transmission (Section 2.4.2). The VBA macro is run for this present study in a Microsoft-Excel environment on an Intel(R) Core(TM) i5-6200U central processing unit (CPU) @ 2.30 GHz with 8.00 GB random access memory (Intel Corporation, Santa Clara, CA, USA). Computation time for 500 cycles of each scenario lies at around 25 min, increasing/decreasing with increasing/decreasing staff size.

## 3. Results and Discussion

The VBA implementation of the hospital model is run for different scenarios of parameter values. The meaningfulness of the study’s results lies not in individual values but the trends which can be observed for parameter variation. Disease parameters or hospital parameters might change for future scenarios. The found effects are valid also for other future settings. The pathogen intake of an organization through pathogen transmission events from patients on to staff, and the spreading of a pathogen between staff must be considered in each healthcare setting.

### 3.1. Base Case and Statistical Convergence

Table 4 shows the model parameters and their values in the base case. The effect of the parameters on e.g., open hospital capacity Γ is discussed in relation to this base case.

Figure 1 shows the output of the model for predicted open hospital capacity, averaged for *n* = 500 runs together with its standard deviation per day for the duration *t* of 100 days. The figure shows that under the top down rollout scheme (blue) a greater decrease in open hospital capacity is expected than under the exposure focused rollout (green). In both cases, the minimum capacity is reached after passing of incubation time and recovery time (15 days). In the beginning, expected open capacity decreases over time as nonvaccinated staff contract the pathogen. This fall does not set in at *t* = 1 day as staff reserves are still available initially. After incubation time and recovery time, the first staff members who contracted the pathogen return to work. The ensuing increase of open hospital capacity is the result of greater immunity of staff to the pathogen due to the vaccine rollout. The fall to the minimum capacity is steeper for the top down rollout, and the following return to full capacity after is more moderate for the top down rollout when compared to the green exposure focused rollout. The predicted better performance of the pathogen focused rollout is independent of the hospital model’s size. It would replicate in an organization employing e.g., 5 or 10 times more staff. A hospital of greater size would have more complex intrastaff transmissions.

Figure 2 shows for predicted hospital capacity in the base case of Table 4 the average and the average’s standard deviation per *n* as developing over *n*. Both rollout schemes converge towards a constant after around *n* = 100. Standard deviation lies below 0.4% and is greater for the top down vaccine rollout (blue). For both rollout schemes, standard deviation converges towards a constant value and changes only negligibly after *n* = 300.

### 3.2. Disease Activity: Influence of Pathogen Infectiousness and Prevalence

Prevalence $p_{pos}$ of the disease in outpatients coming into A&E is the main variable determining pathogen intake into the hospital. Infectiousness $R_{S}$ decides about the number of transmission events from patients onto staff, or between staff. To investigate the effect of increased disease spreading, a parameter study is performed where the parameters $R_{S}$ and $p_{pos}$ are varied according to the values given in Table 5 while all other parameters are kept as described in the model’s base case scenario of Table 4. $R_{S}$ is in the magnitude as obtained in [31] for Sars-Cov-2. $p_{pos}$ is varied between 0.1 and 1.

Figure 3 shows the capacity for both rollout schemes for all resulting $p_{pos}*R_{S}$. The model predicts that for greater disease activity the advantage of the exposure focused rollout increases until a nearly constant level is reached at around $p_{pos}*R_{S}=$ 5. At this constant level, predicted open capacity under the exposure focused rollout is about 1.2 times greater than what is predicted for the top down rollout (orange). When disease activity decreases, the advantage of the exposure focused rollout does so as well. Both rollout schemes achieve approximately equal results for $lim_{\left(p_{pos}*R_{S}\right)}\to 0^{+}$.

Figure 4 gives information about the influence of disease activity on open status on unit level. The base case scenario (*R_S_* = 7, *p_pos_* = 0.25, Table 4) leads to a temporary total closure of ward 3 and substantial closures of ward 2 and ward 1 under the top down rollout. The prediction for exposure focused vaccine rollout of the base case predicted a fully open ward 1. Ward 2 and 3 would be expected to lose temporarily capacity, by a margin less than what is predicted for the top down scheme. If disease activity is increased to e.g., (*R_S_* = 10, *p_pos_* = 0.75) under both rollout schemes open capacity drops. Temporarily, ward 1–3 are predicted closed and A&E predicted not fully open under the top down rollout while A&E is predicted fully open under the exposure focused rollout. Also, the predicted disease spreading pattern between hospital wards is independent of the total staff number. Larger healthcare organizations, be they hospitals or nursing homes, can have multiple subunits between which patients and staff are moving during the day. For modelling that setting, the number of infected, nonsymptomatic colleagues $\bar{sym}$ (please see Equations (2) and (3) can be expanded. Irrespectively, the pathogen intake of larger multiclinic hospital organizations will be as modelled here.

### 3.3. Influence of Recovery Time and Social Considerations

The time needed by staff to recover *t_rec_* [d] from the simulated pathogen can vary based on the pathogen and predisposition of a staff member. To quantify the influence on the model’s prediction, *t_rec_* is simulated for the values shown in Table 6 while setting all other parameters to the base case of Table 4. Figure 5 shows the obtained results, formatted as before Figure 3 (exposure focused rollout green, top down rollout blue, ratio of both schemes orange).

Figure 5 shows that expected open hospital capacity decreases for greater *t_rec_* of infected staff. The advantage of the exposure focused rollout increases for greater *t_rec_*. Similarly to Figure 3 for disease activity, the difference factor (orange) lies at around 1.2 for the maximum value simulated.

Figure 6 gives the expected number of infected staff over time, top down rollout scheme subtracted from exposure focused rollout. Total numbers are shown by hierarchy group. This analysis is of importance as it relates to work which showed that social factors must be considered during the current pandemic [24,26,27]. The greatest difference between the vaccine rollout schemes is seen for the nursing staff. Their infection numbers are predicted by the model to increase most, once the hierarchical vaccine allocation takes place. Executive seniors and senior doctors are predicted to on average benefit from the top down rollout. The chief who also can work on A&E is vaccinated fairly early in both schemes and not predicted to contract the pathogen. Interestingly, the physicians are treated favorably in each of the two rollout schemes. Depending on progression of the rollout, they benefit or not. Initially, they are worse off as doctors of greater hierarchical rank are assigned vaccine doses. After *t* of 27 days, their group however benefits from the top down scheme as they are assigned vaccine doses which would go alternatively to A&E nurses under the exposure focused rollout. Total staff size of a hospital does not alter this effect. In considerations for a vaccine rollout in private practices, this finding is of importance as nurses there typically see the same or even greater pathogen exposure than the doctor.

### 3.4. Availability of Vaccine and Rollout Rate

A very urgent aspect of the current rollout is the scarcity of vaccine doses as a resource. Different vaccines have been approved by regulatory bodies, more are in the final stage of certification [1,2]. However, production numbers do not yet provide sufficient supply for an immediate full rollout. In today’s real-world situation, vaccines are rationed [25] initially also for smaller hospitals. The model predicts for the two rollout schemes that the difference in expected open hospital capacity increases for greater resource scarcity, Figure 7 and Table 7. This effect is independent of hospital model size.

The availability of vaccine doses *v_S_* [1/d] is iterated for the values of Table 7 while keeping all other parameters as specified for the base case Table 4. Figure 7 shows that the maximum of the relative comparison between both schemes (orange) is located at 1/d. For the staff size as specified in Table 1, no measurable difference exists in the model once supply of *v_S_* > 5/d is available. This means that in particular in situations of resource scarcity, the exposure focused rollout scheme is advantageous when measured in open hospital capacity.

### 3.5. Influence of Staff Reserve

Another resource in hospital operations, apart from the vaccines per day available for the hospital, is the staff. The model calculates a partial or full closure of units as penalty condition for staff shortage due to disease, Table 2 and Table 8. For the variation of initial staff reserve, the model predicts that in each scenario the exposure focused rollout scheme leads to greater expected open hospital capacity.

Figure 8 shows the numbers obtained for the variation of staff reserve. Similarly as before in Figure 7 for the availability of vaccine doses, the difference obtained for exposure focused and top down rollout is greater when base staffing or only reduced staffing is initially available. With increased staffing, the difference between the two rollout schemes decreases. Here again, results can be interpreted in a way that under shortage of resources the exposure focused vaccine rollout is more vital for the hospital’s ability to operate. Here, staff size has an influence on simulation results in the form that reserve is beneficial. Staff reserve is beneficial independently of total staff number.

### 3.6. Consideration of Increased Transmission due to Age and Roles in Hospital Hierarchy

In the argumentation for the case of the top down rollout scheme, the special importance of higher staff ranks for hospital operations is one factor. On top, their typically greater age makes them more susceptible for pathogen transmission. Their exposure to patients is typically reduced compared to nurses and junior doctors. Higher ranks are involved in administrative tasks which during the pandemic can partially even be done in home office. The model is iterated over *T_S_* [-] (increased pathogen transmission probability for greater age) and *H_S_* [-] (reduced patient contact for the hospital’s chief and executive seniors) to demonstrate the influence of both variables. Values are shown in Table 9. Both are varied between 1 and 20.

Figure 9 shows the predicted open hospital capacity of the base case (Table 4) for the variation of age dependency of pathogen transmission probability, *T_S_*. Under both rollout schemes, expected open capacity decreases for greater age factor. The model predicts better outcome for the exposure focused rollout (orange). This means that the influence of age (distribution per hierarchy group in Table 3) does not serve as an argument for the case of the top down rollout scheme but rather against it. It must not be forgotten that the age factor applies to all hospital staff. Nurses realistically also reach age of 60 and over, like the hospital’s chief when they are equally more probable to contract the pathogen. An increase of model size would not change the received results pattern. The age distribution defined for the current study defines a typical age structure found around the world, including junior doctors up to hospital chiefs.

The influence of *H_S_* is found interestingly to be negligible in the hospital model as defined in the present study. No clear influence on expected open hospital capacity or on infection numbers amongst the chief and executive seniors is predicted in the per cent or per mill range. Increasing the number of statistical cycles to cycle numbers greater than the applied *n* = 500 is possible. It isn’t performed as part of the present study however as it would generate numerical results which suggest pseudoprecision that is beyond the predictive precision of the hospital model. Total staff number does not change the obtained pattern.

This lack of explanatory power of the hospital model with regard to *H_S_* is explicable by the rollout schemes as defined under Section 2.3. The chief and executive seniors are first to be vaccinated for the top down scheme. In the exposure focused rollout scheme, the chief or the executive senior on A&E shift is the first person of the entire staff to be vaccinated at *t* = 1 day. Under this setting, exposure to the nonvaccinated chief or executive seniors is minimal already and the influence of *H_S_* does not reduce substantially further the probability of pathogen transmission.

## 4. Conclusions and Future Work

The model’s implementation in VBA is iterated for parameter studies. The two rollout schemes for a top down or exposure focused assignment of vaccine doses to staff can be compared in various settings.

In this hospital model, the expected open hospital capacity which is assumed to maximize patient benefit is generally greater for the exposure focused rollout. The model relies on the drawing of random numbers for e.g., age and transmission events. Building statistical averages for *n* = 500 model runs ensures convergence of results with a constant, Figure 2. Standard deviation of open hospital capacity calculated over averages of *n* lies below 0.4%.

Results predict an increasing advantage by the exposure focused vaccine rollout scheme under greater disease activity defined by greater pathogen infectiousness and prevalence in patients, Figure 3. Overall, open hospital capacity can be broken down in open status of hospital units (A&E, wards 1–3). Trends observable for total capacity translate into effects on unit level. Following unit prioritization, predicted loss of unit open status is more pronounced under the top down rollout scheme. In addition, an increasing recovery time leads to greater advantage of the exposure focused rollout, Figure 5.

The results in Figure 6 demonstrate the different infection numbers per hierarchy groups. The top down rollout scheme would be most disadvantageous for nurses. Their infection numbers increase greatly under that scheme. In addition, nurses on A&E get vaccinated only after all doctors have been assigned doses. These comparisons between prioritization schemes are of importance. It is known that social status, or minority status influences disease spreading [24,26,27].

Shortage of resources (supply of vaccine doses, or initial staff reserve) decreases the predicted open hospital capacity. The exposure focused rollout scheme leads to better results in those scenarios and is the preferable option when compared to the top down rollout scheme, Figure 7 and Figure 8. The influence of greater pathogen transmission probability by greater age is another argument for the case of the exposure focused rollout scheme. Here again, the role of nursing staff who realistically reach ages of 60 and over must be considered.

The implemented and presented model is a first step to investigate decision making strategies for a novel problem [22,23]. The model’s complexity is kept to a minimum. Several other ongoing hospital processes could be implemented in it. Extensions could investigate effects of staff moving between hospitals, only partial immunity of staff after a first vaccine dose, or the fact that tests produce false negative and false positive results. The organizational structure of the hospital could be extended so that it contains several medical specialties; surgery, medicine, radiology, and/or anesthesia. For a hospital’s ability to perform presurgical imaging diagnostics and surgery itself, staffing in radiology and in anesthesia is essential.

The foremost limitation of the simulated model is the small size of the simulated unit. Larger organizations can be expected to have exchange between clinics with pathogen transmission events. This is true for staff who might work on wards shared by clinics; and for patients who receive treatment from more than only one clinic. Disease spreading will be more difficult to simulate. The effects from vaccine rollout schemes will be more difficult to isolate and measure. Nonetheless, observed effects in this study are transferable also to larger organizations. The shown influence on nursing staff, the effect of staff age, or the beneficial effect of greater staff reserves will equally be valid; independently of hospital size. Hospital structure with staff ranks, staff numbers, and staff age reflects a system typical for hospitals worldwide. Changes within the defined staff pyramid will not alter the obtained results; overall results patterns are independent of this. Other healthcare settings can similarly be investigated for optimized vaccine rollout schemes. Healthcare workers in private practice are typically enter into contact with outpatients who can’t be tested in advance before entering the premises. Nursing homes, however, might have the resources to isolate new arriving inpatients for 24–48 h (today the typical waiting time for the SARS-CoV-2 test by polymerase chain reaction). If negative, the new inpatient can be assumed noninfectious, and he or she leaves isolation. Home care services on the other hand will be in touch with patients of unknown disease status.

The mathematical evaluation in the presented model implementation builds averages over time. In a more complex implementation of a more diversified hospital model, integrating open status over time and comparison of areas (closed/open) can increase explanatory power.

The model so far only quantifies impact on unit open status. This could be extended into calculations about number of procedures and number of patients affected. The number of patients treated and remuneration by diagnosis-related groups would allow a quantification of the financial impact on annual business results of the hospital.

The scientific understanding of SARS-CoV-2 behavior is expected to increase. Based on this, SARS-Cov-2 mutants, and other future scenarios, new vaccine rollouts might become necessary. After four major outbreaks of infectious diseases in e.g., Hong Kong alone during the 25 years before the coronavirus pandemic [35], similar situations might repeat in the future.

## Figures and Tables

**Figure 1 vaccines-09-00546-f001:**
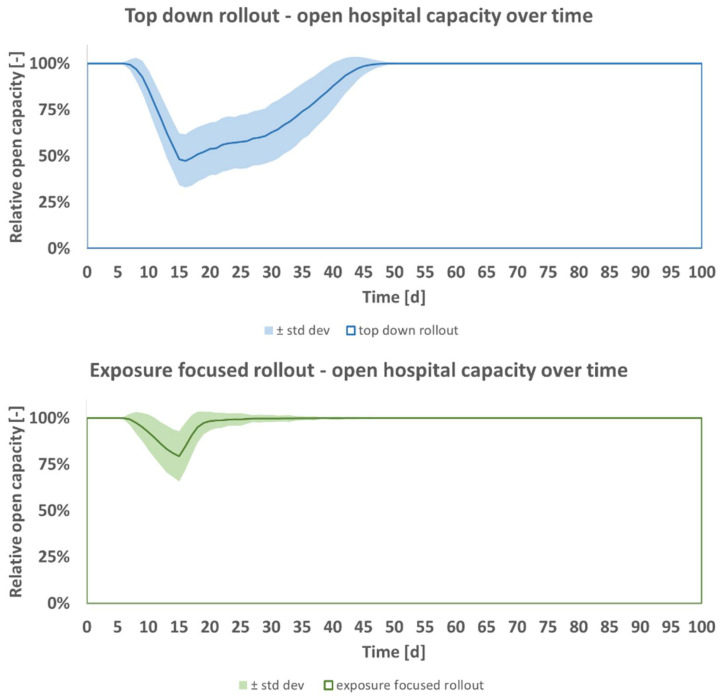
Open hospital capacity, predicted mean ± standard deviation per day, for top down (blue) and exposure focused (green) vaccine rollout, base case scenario of Table 4.

**Figure 2 vaccines-09-00546-f002:**
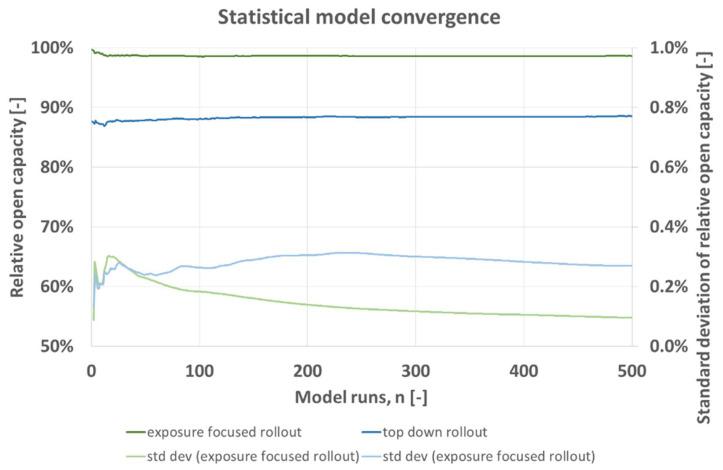
Statistical model convergence, expected relative open capacity averaged for *n* runs with corresponding standard deviation per model run *n*, for top down (blue) and exposure focused (green) vaccine rollout, base case scenario of Table 4.

**Figure 3 vaccines-09-00546-f003:**
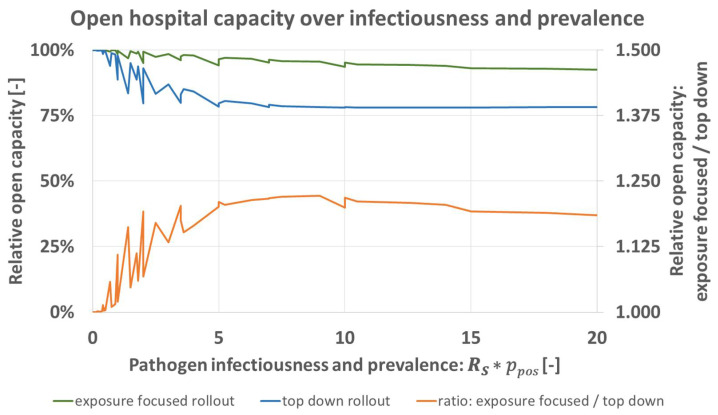
Predicted open hospital capacity over product of infectiousness *R_S_* and prevalence in A&E-patients *p_pos_* for top down (blue), and exposure focused rollout (green), together with their respective ratio (orange).

**Figure 4 vaccines-09-00546-f004:**
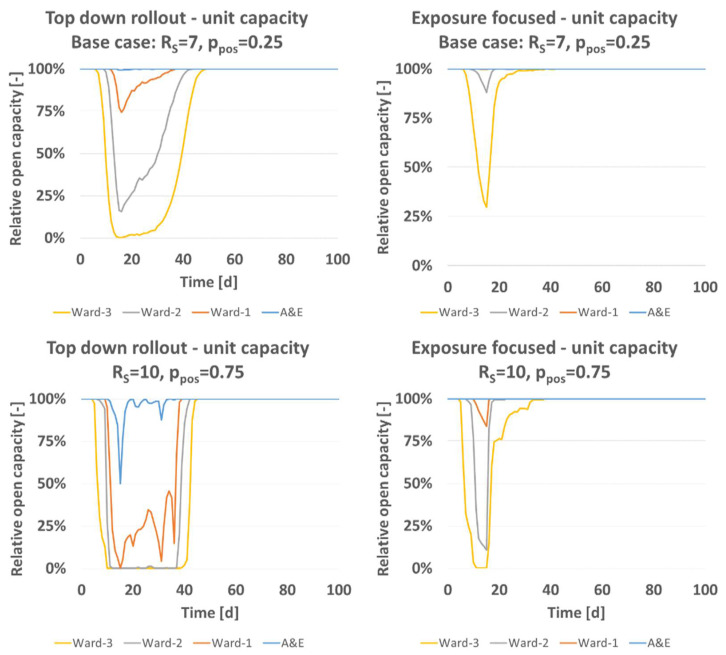
Relative open capacity split on to unit level for base case (*R_S_* = 7, *p_pos_* = 0.25, Table 4), and parameter variation (*R_S_* = 10, *p_pos_* = 0.75).

**Figure 5 vaccines-09-00546-f005:**
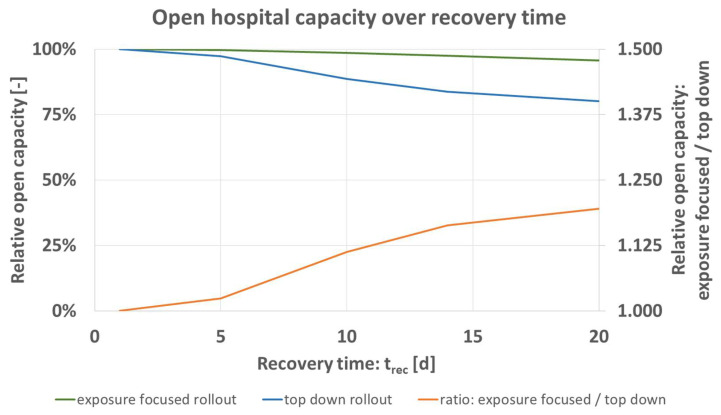
Predicted open hospital capacity over recovery time *t_rec_* [d] for top down (blue), and exposure focused rollout (green), together with their respective ratio (orange).

**Figure 6 vaccines-09-00546-f006:**
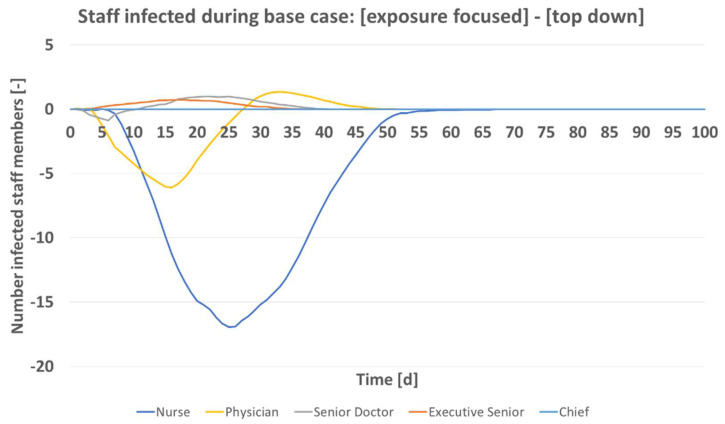
Predicted number of infected staff by hierarchy group (Table 1) over time for base case scenario (Table 4).

**Figure 7 vaccines-09-00546-f007:**
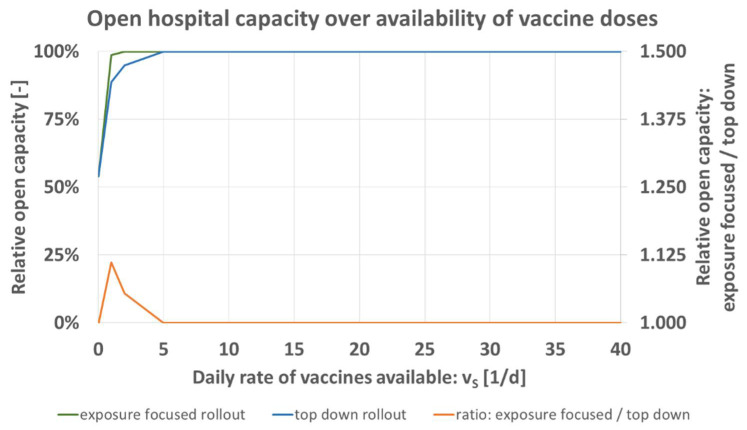
Predicted open hospital capacity over daily rate of vaccines available *v_S_* [1/d] for top down (blue), and exposure focused rollout (green), together with their respective ratio (orange).

**Figure 8 vaccines-09-00546-f008:**
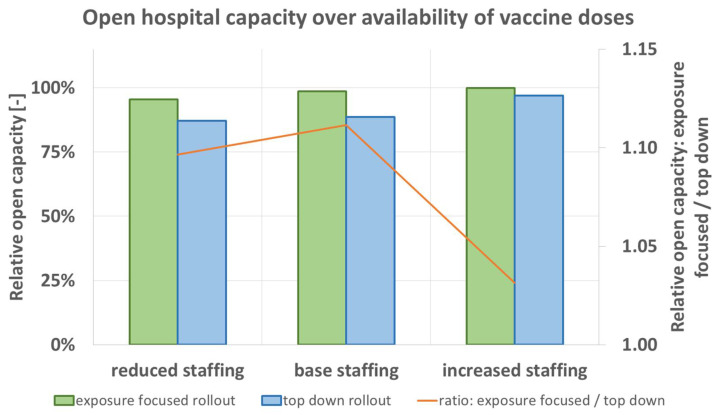
Predicted open hospital capacity depending on initial staff reserve (Table 1) for top down (blue), and exposure focused rollout (green), together with their respective ratio (orange).

**Figure 9 vaccines-09-00546-f009:**
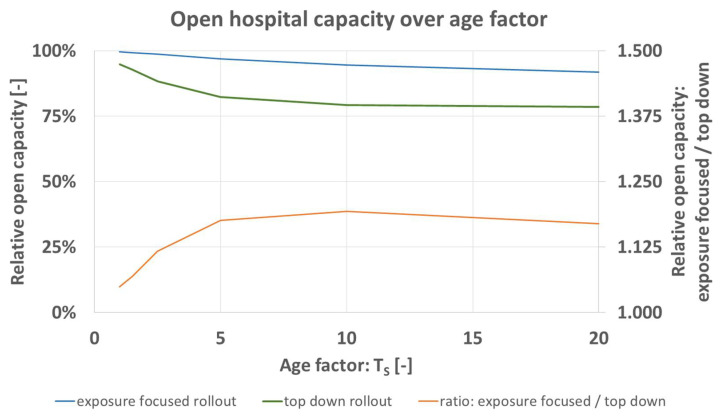
Predicted open hospital capacity over age factor *T_S_* [-] for top down (blue), and exposure focused rollout (green), together with their respective ratio (orange).

**Table 1 vaccines-09-00546-t001:** Overview of staffing scenarios.

Rank	Base Staffing		Reduced Staffing		Increased Staffing	
	Total	Reserve	Total	Reserve	Total	Reserve
Chief	1	Combined 2	1	Combined 0	1	Combined 4
Executive Senior	5		3		7	
Senior Doctor	6	2	5	1	8	4
Physician	14	4	12	2	18	8
Nurse	33	8	29	4	41	16

**Table 2 vaccines-09-00546-t002:** Staff requirement for full or partial unit operations.

Rank	A&E		Ward 1 to 3	
	100%	50%	100%	50%
Chief	Combined 1	Combined > 0	Combined 1	Combined > 0
Executive Senior				
Senior Doctor	1		1	
Physician	4	>1	2	>0
Nurse	10	>4	5	>2

**Table 3 vaccines-09-00546-t003:** Lower bound and upper bound for age distribution.

Rank	Lower Bound	Upper Bound
Chief	51	65
Executive Senior	41	50
Senior Doctor	33	40
Physician	25	32
Nurse	18	65

**Table 4 vaccines-09-00546-t004:** Parameters of the base case scenario.

Parameter	*v_S_* [1/d]	*R_S_* [-]	*p_pos_* [-]	*t_inc_* [d]	*t_rec_* [d]	*T_S_* [-]	*H_S_* [-]	Staffing
Value	1	7	0.25	5	10	2.5	5	Base staffing

**Table 5 vaccines-09-00546-t005:** Variation of infectiousness and exposure.

Parameter	Simulated Values
*R_S_* [-]	[0.1, 0.5, 1, 2, 4, 7, 10, 14, 20]
*p_pos_* [-]	[0.1, 0.25, 0.5, 0.75, 0.9, 1]

**Table 6 vaccines-09-00546-t006:** Variation of recovery time.

Parameter	Simulated Values
*t_rec_* [d]	[1, 5, 10, 14, 20]

**Table 7 vaccines-09-00546-t007:** Variation of daily rate of vaccines.

Parameter	Simulated Values
*v_S_* [1/d]	[0, 1, 2, 5, 7, 10, 20, 30, 40]

**Table 8 vaccines-09-00546-t008:** Variation of staff size and initial staff reserve, numbers defined in Table 1.

Parameter	Simulated Values
Staffing	[Base staffing, Reduced staffing, Increased staffing]

**Table 9 vaccines-09-00546-t009:** Variation of age factor and reduced exposure due to hierarchy.

Parameter	Simulated Values
*T_S_* [-]	[1, 1.5, 2.5, 5, 10, 20]
*H_S_* [-]	[1, 2, 5, 10, 20]

## Data Availability

The data presented in this study is available open access as macro.vba and as macro.xlsm at doi.org/10.5281/zenodo.4589332.

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
