# Peer review of "Maximization of Open Hospital Capacity under Shortage of SARS-CoV-2 Vaccines—An Open Access, Stochastic Simulation Tool"

_vaccines, 2021, doi:10.3390/vaccines9060546_

Round 1

Reviewer 1 Report

This is an interesting manuscript that develops a model to determine the most effective vaccine rollout strategies in a defined type of hospital. The main problem with this paper is that describes a situation that is very concrete and I have serious doubts about its generalizability outside the very concrete parameters authors have considered. Their assumptions are based on a very small-size hospital, with a very limited number of healthcare workers. In a hospital this size I don´t think that even this modeling may have any role in decision-making: the number of professionals is so low that you would be able to vaccinate all of them in 1 single day. In larger hospitals, the situation may be different, but I´m not sure that this model may be applied in other settings. In any case, technical expertise in reviewing the model specifications would be necessary.

Author Response

Comments reviewer #1:

We would like to thank the reviewer for providing insightful comments on our paper, which helped us to improve the manuscript. In what follows we address the reviewer’s comments

“This is an interesting manuscript that develops a model to determine the most effective vaccine rollout strategies in a defined type of hospital. The main problem with this paper is that describes a situation that is very concrete and I have serious doubts about its generalizability outside the very concrete parameters authors have considered. Their assumptions are based on a very small-size hospital, with a very limited number of healthcare workers. In a hospital this size I don´t think that even this modeling may have any role in decision-making: the number of professionals is so low that you would be able to vaccinate all of them in 1 single day. In larger hospitals, the situation may be different, but I´m not sure that this model may be applied in other settings. In any case, technical expertise in reviewing the model specifications would be necessary.”

The reviewer is correct in pointing out that the simulation results are valid only for the specified model parameters, model simplifications, and modelling assumptions. Also, he is right in pointing out that conclusions from the presented model on future real-world scenarios must be interpreted within these limitations. We have inserted a comment on results interpretation in section 1 (line 84-86).

However, as we point out in the revision, we are confident that the results can be transferred onto larger hospitals. In the revised paper, we have added the following in section 4 (line 431-439):

“The foremost limitation of the simulated model is its limited size. Larger organizations can be expected to have exchange between clinics with pathogen transmission events. This is true for staff who might work on wards shared by clinics; and for patients who receive treatment from more than only one clinic. Disease spreading will be more difficult to simulate. The effects from vaccine rollout schemes will be more difficult to isolate and measure. Nonetheless, observed effects in this study are transferable also to larger organizations. The shown influence on nursing staff, the effect of staff age, or the beneficial effect of greater staff reserves will equally be valid; independently of hospital size.

Reviewer 2 Report

This is an interesting study that aimed to investigate consequences from different rollout schemes. The study focused on the maximisation of open hospital capacity which is assumed to maximise patient benefit. English language should be carefully revised by a native speaker; I also suggest to check throughout the text for spelling errors and consistent use of abbreviations. The corresponding author should be indicated. The tables and figures are adequate. Strengths and limitations should be clearly explained in the text. I would suggest to include further discussion on the future direction and possible clinical application of the results.

Author Response

Comments reviewer #2:

“This is an interesting study that aimed to investigate consequences from different rollout schemes. The study focused on the maximisation of open hospital capacity which is assumed to maximise patient benefit.”

We would like to thank the reviewer for his positive assessment of our work.

“English language should be carefully revised by a native speaker; I also suggest to check throughout the text for spelling errors and consistent use of abbreviations. The corresponding author should be indicated. The tables and figures are adequate.”

We have reworked the manuscript and edited it to be American spelling.

Abbreviations are introduced at first instance, now also:

  • GNU's Not Unix (GNU), line 74
  • random access memory (RAM), line 206
  • central processing unit (CPU), line 205-206

The corresponding author is highlighted and email address added.

“Strengths and limitations should be clearly explained in the text. I would suggest to include further discussion on the future direction and possible clinical application of the results.”

Conclusions and discussion in the text are extended as suggested also by reviewer #1. We discuss in greater detail limitations and strengths of the study, in particular predictive possibilities from the presented model, section 4 (line 431-439):

“The foremost limitation of the simulated model is its limited size. Larger organizations can be expected to have exchange between clinics with pathogen transmission events. This is true for staff who might work on wards shared by clinics; and for patients who receive treatment from more than only one clinic. Disease spreading will be more difficult to simulate. The effects from vaccine rollout schemes will be more difficult to isolate and measure. Nonetheless, observed effects in this study are transferable also to larger organizations. The shown influence on nursing staff, the effect of staff age, or the beneficial effect of greater staff reserves will equally be valid; independently of hospital size.

Round 2

Reviewer 1 Report

The COVID-19 pandemic has put healthcare systems under unprecedented stress to accommodate unexpected numbers of patients forcing a quick re-organization. This manuscript describes a model for rolling out vaccination to maintain hospital capacity during epidemic outbreaks. The main limitation of this work, which was already indicated and that authors have not improved, is that the authors have developed their model for one hospital, small in size and with very a specific human resources structure. It is really difficult to see that readers working in other healthcare settings may find this model useful if what they need is to generate rapid, locally valid responses to unexpected infectious disease epidemic challenges.

Author Response

Comments reviewer #1:

We would like to thank the reviewer #1 again for taking the time to read our paper and to provide his insights. In what follows, we state how we have addressed the reviewer’s comments. We hope that this revision settles all concern of the reviewer.

“The COVID-19 pandemic has put healthcare systems under unprecedented stress to accommodate unexpected numbers of patients forcing a quick re-organization. This manuscript describes a model for rolling out vaccination to maintain hospital capacity during epidemic outbreaks.

The main limitation of this work, which was already indicated and that authors have not improved, is that the authors have developed their model for one hospital, small in size and with very a specific human resources structure.”

We have added further information about the size of the simulation and the staff structure in the manuscript.

We have added in the introduction (line 82-87):

The simulated hospital and affected staff are specified in Sections 2.1, and 2.2. Staff structure is a pyramid. This hierarchy form is found worldwide as typical human resources (HR) structure in hospitals. Staff further up in the hierarchy is of greater age. Larger simulations are possible in the future. The investigated effects are independent of hospital size. In Section 4, we discuss observable trends and predictions for future scenarios.

For methodology, we have added (line 94-97):

For the present study, a hypothetical hospital is modelled as it can be found around the world. Staff on higher hierarchy levels are fewer in numbers, of greater age, and of greater individual importance for the functioning of the hospital organization. Influence of model parameters is discussed in the results section and in the conclusions section.

To the paragraph about Table a, we have added (line 100-101):

Staff size and staff requirements are chosen to reflect a typical hospital situation.

To the paragraph about Table b, we have added (line 116-118)

The specified staff requirements reflect a typical contemporary hospital organization. The total and relative numbers are chosen to imitate a general medicine or surgery clinic.

To the paragraph about Table c, we have added (line 128-130)

Age boundaries vary between different countries; e.g. there is no global uniform retirement age. The chosen age distribution reflects a typical hospital setting of our time.

In the results section, about the findings of Fig.1 we have added about hospital size (line 248-251):

The predicted better performance of the pathogen focused rollout is independent of the hospital model’s size. It would replicate in an organization employing e.g. 5 or 10 times more staff. A hospital of greater size would have more complex intra-staff transmissions.

The explanations about Fig. 4 are extended by the following with regard to pathogen intake and intra-staff transmission events, with special consideration of multi-unit healthcare organizations (line 294-300):

Also, the predicted disease spreading pattern between hospital wards is independent of the total staff number. Larger healthcare organizations, let it be hospitals or nursing homes, can have multiple subunits between which patients and staff are moving during the day. For modelling that setting, the number of infected, non-symptomatic colleagues ??? (please see Eq. 2 and 3) can be expanded. Irrespectively, the pathogen intake of larger multi-clinic hospital organizations will be as modelled here.

The effect on nursing staff is discussed also with regard to total staff number (line 334-337):

Total staff size of a hospital does not alter this effect. In considerations for a vaccine rollout in private practices, this finding is of importance as there nurses typically see same or even greater pathogen exposure than the doctor.

It is added for the effect of resource scarcity (line 349):

This effect is independent of hospital model size.

Staff reserve and its effect are discussed with regard to total staff number (line 377-378):

Here, staff size has an influence on simulation results in the form that reserve is beneficial. Staff reserve is beneficial independently of total staff number.

We have added on the effect of staff age (line 402-405):

An increase of model size would not change the received results pattern. The age distribution defined for the current study defines a typical age structure found around the world; including junior doctors up to hospital chiefs.

The effect of the hierarchy factor is independent of total staff number (line 415-416):

Total staff number does not change the obtained pattern.

“It is really difficult to see that readers working

in other healthcare settings may find this model useful if what they need is to generate rapid, locally valid responses to unexpected infectious disease epidemic challenges."

Our work relates to the settings as pointed out in the introduction [22,23]. We have however used this as an opportunity to extend the manuscript in conclusions to address other healthcare settings as well. About pathogen transmission in other healthcare settings we have added in methodology (line 167-169):

Other non-hospital healthcare settings as found in private practice, nursing homes, or at home care services can be simulated alike, based on their specific pathogen exposure patterns.

In results, we have added (line 227-229):

The pathogen intake of an organization through pathogen transmission events from patients onto staff, and the spreading of a pathogen between staff must be considered in each healthcare setting.

In the outlook we also now mention other non-hospital healthcare settings (line 472-481):

Hospital structure with staff ranks, staff numbers, and staff age reflects a system typical for hospitals worldwide. Changes within the defined staff pyramid will not alter the obtained results; overall results patterns are independent of this. Other healthcare settings can similarly be investigated for optimized vaccine rollout schemes. Healthcare workers in private practice are typically coming in touch with outpatients who can’t be tested in advance before entering the premises. Nursing homes however might have the resources to isolate new arriving inpatients for 24-28 hs (today the typical waiting time for the SARS-CoV-2 test by polymerase chain reaction). If negative, the new inpatient can be assumed non-infectious, and he or she leaves isolation. Home care services on the other hand will be in touch with patients of unknown disease status.

We were hoping to make a valuable contribution to the topic of our specified research question. Could the reviewer maybe provide here more insight into what is looking for in the manuscript if the reviewer disagrees?

Reviewer 2 Report

The paper has been improved. The authors replied satisfactorily to all my comments.

Round 3

Reviewer 1 Report

I would suggest including in the title that this is an application of a general model to a specific hospital size-structure

Author Response

Comments reviewer #1:

“I would suggest including in the title that this is an application of a general model to a specific hospital size-structure.”

We would like to thank the reviewer #1 for reading our edited manuscript.

We have edited the submission’s title as follows:

Maximization of open hospital capacity under shortage of SARS-CoV-2 vaccines - an open access, stochastic simulation tool